# Research on Wind Turbine Fault Detection Based on the Fusion of ASL-CatBoost and TtRSA

**DOI:** 10.3390/s23156741

**Published:** 2023-07-28

**Authors:** Lingchao Kong, Hongtao Liang, Guozhu Liu, Shuo Liu

**Affiliations:** School of Information Science and Technology, Qingdao University of Science and Technology, Qingdao 266061, China; kk1392567492@163.com (L.K.); lgz_0228@163.com (G.L.); ls354304274@163.com (S.L.)

**Keywords:** wind turbine, CatBoost algorithm, fault detection, category imbalance, intelligent optimization algorithm

## Abstract

The internal structure of wind turbines is intricate and precise, although the challenging working conditions often give rise to various operational faults. This study aims to address the limitations of traditional machine learning algorithms in wind turbine fault detection and the imbalance of positive and negative samples in the fault detection dataset. To achieve the real-time detection of wind turbine group faults and to capture wind turbine fault state information, an enhanced ASL-CatBoost algorithm is proposed. Additionally, a crawling animal search algorithm that incorporates the Tent chaotic mapping and t-distribution mutation strategy is introduced to assess the sensitivity of the ASL-CatBoost algorithm toward hyperparameters and the difficulty of manual hyperparameter setting. The effectiveness of the proposed hyperparameter optimization strategy, termed the TtRSA algorithm, is demonstrated through a comparison of traditional intelligent optimization algorithms using 11 benchmark test functions. When applied to the hyperparameter optimization of the ASL-CatBoost algorithm, the TtRSA-ASL-CatBoost algorithm exhibits notable enhancements in accuracy, recall, and other performance measures compared with the ASL-CatBoost algorithm and other ensemble learning algorithms. The experimental results affirm that the proposed algorithm model improvement strategy effectively enhances the wind turbine fault detection classification recognition rate.

## 1. Introduction

With the rapid development of the global economy, the scale of demand for energy continues to expand. Traditional thermal power generation methods are highly prone to causing environmental pollution and they do not meet the requirements of sustainable development. Green power generation methods such as wind power generation are in line with the future development direction of the energy industry. With the development of Industry 4.0, the global installed capacity of wind turbines is expected to reach two billion kilowatts by 2030 [1]. Promoting the development of renewable energy will not only meet the energy needs of economic development, but also reduce the proportion of traditional thermal power generation methods, and accelerate the construction of a clean, low-carbon, energy-efficient system [2]. However, with the continuous expansion of wind power generation, and an increase in operational lifespan, the issue of turbine faults has become increasingly prominent, posing a series of challenges to the wind power industry. The conventional maintenance approach for wind turbine units is typically based on scheduled inspections and fault responses, which presents several issues. Firstly, scheduled inspections often fail to accurately predict the occurrence time and types of turbine faults, which can result in unnecessary maintenance and high costs. Secondly, maintenance carried out in response to faults is often conducted after the occurrence of the faults, potentially leading to the prolonged downtime of the units and a reduction in production capacity. Additionally, the remote geographical locations where wind turbine units are typically installed pose significant challenges and high costs for maintenance, limiting their reliability and maintainability. Therefore, the development and implementation of wind turbine fault detection systems are of great significance. By leveraging advanced sensor technologies, data analysis, and machine learning algorithms, real-time monitoring of the operational status and performance parameters of the units can be achieved, enabling the timely detection of potential faults and abnormalities [3]. Accurate fault detection can help reduce maintenance costs, improve maintenance efficiency, and optimize operations and equipment performance [4,5]. Moreover, by enhancing the accuracy and timeliness of fault detection, the safety of wind turbine units can be enhanced, reducing the risk of accidents and promoting the overall sustainable development of the wind power industry.

Artificial intelligence and machine learning have unique advantages in the field of wind turbine fault detection [6,7,8,9]. In order to improve the detection speed and accuracy of wind turbine faults, a novel dynamic model sensor method is proposed for SCADA data-based wind turbine fault detection. A dynamic model representing the relationship between the generator temperature, wind speed, and ambient temperature is constructed using the first principles, and it is used as the basic structure of the model sensor. When the model sensor is applied for fault detection, its parameters are updated regularly using the generator temperature, wind speed, and ambient temperature data from the SCADA system. Then, from the updated model, the fault sensitive features of the wind turbine system are extracted by performing system frequency analysis for use in turbine fault detection [10]. Aziz, U [11] used a realistic framework for SCADA data simulation by critically comparing power-based wind turbine fault-detection methods. Song [12] proposed the use of an improved denoising autoencoder to detect wind turbine rolling bearing faults. Liu [13] proposed a twin neural network method that allows the algorithm to achieve wind turbine fault detection with only a small amount of training data. In order to solve the problem of inaccurate and untimely fault detection caused by wind turbine data features, Liu [14] proposed a new deep network called the Deep Residual Network (DRN) for the fault detection of wind turbines. The results indicate that the proposed DRN achieves a better performance and outperforms some published fault detection methods. However, none of the aforementioned studies considered the impact of the imbalanced positive and negative samples in wind turbine fault detection datasets on the accuracy of algorithms, leaving room for further improvement in the field concerning the detection accuracy of faults.

Fault detection models typically contain many hyperparameters that are not learned from data, but are manually set by the user. These hyperparameters play a crucial role in determining the performance and behavior of the model. However, finding the optimal combination of hyperparameters can be a challenging and time-consuming task. Swarm intelligence optimization algorithms can search for the optimal solution distributed in a certain range of space, and they have good parallelism and autonomous exploration, which is of significance for the hyperparameter optimization of fault detection models [15,16,17]. Lei [18] analyzed a fault detection model based on long short-term neural networks and Bayesian optimization algorithms, and they applied the model to the fault warning of the induced draft fan of a coal-fired power plant, achieving good results. Zhang [19] used the Whale Optimization Algorithm in order to find the global optimal solution, realize hyperparameter optimization of the BiLSTM network, improve the prediction accuracy of wind power generation, and save a significant amount of time on debugging. Huang [20] proposed a crawling animal search algorithm based on the interactive cross strategy of Levy flight, and they verified its effectiveness in practical engineering via innovations such as the welding beam design.

Based on the above research results and their shortcomings, the following work was conducted in this paper. Firstly, the accuracy and recall rate of the CatBoost algorithm to achieve fault detection is not high enough, and there is an imbalance between positive and negative samples in the wind turbine fault detection dataset. The cross-entropy loss function of the CatBoost algorithm was thus replaced by an asymmetric loss function, and the ASL-CatBoost algorithm was proposed to implement wind turbine icing fault detection and capture the state information of wind turbine faults. To verify the effectiveness of the improved method, experiments were conducted using integrated learning algorithms (CatBoost, XGBoost, LightGBM, etc.), machine learning algorithms (SVM), and deep learning algorithms (LSTMAE) as comparison algorithms for the wind turbine icing fault dataset. The experimental results show the superior performance of the improved algorithms. Secondly, due to the difficulty in setting appropriate hyperparameters in the ASL-CatBoost algorithm, in this paper, we propose an improved reptile search algorithm based on the tent chaotic map and t-distribution mutation strategy to optimize hyperparameters such as learning rate, iteration number, and the tree depth of the ASL-CatBoost algorithm. The improved algorithm was also proposed in order to introduce the optimized hyperparameters into the ASL-CatBoost algorithm for model training. To verify the optimization ability of the improved reptile search algorithm, the TtRSA algorithm was compared with classical population intelligence optimization algorithms such as PSO, WOA, and SSA regarding 11 benchmark functions. The experimental results show that the improved reptile search algorithm has better performance, convergence speed, and accuracy. Finally, the TtRSA-optimized ASL-CatBoost algorithm has a higher detection accuracy and detection efficiency than the original ASL-CatBoost algorithm.

## 2. Materials and Methods

### 2.1. CatBoost Algorithm

In 2017, Yandex proposed a new integrated learning algorithm called CatBoost [21]. This algorithm is an improvement on Gradient Boosting Decision Trees, and it outperforms other algorithms in the same GBDT [22] framework, such as XGBoost [23] and LightGBM [24], in terms of model accuracy. The main innovation of the CatBoost algorithm is the use of Ordered Boosting instead of the traditional gradient estimation method, which solves the problems of Gradient Bias and Prediction Shift. In addition, the algorithm uses Oblivious trees as base models, which improves the model’s ability to classify correctly, and it takes into account its generalization ability, effectively preventing algorithm over fitting.

The Gradient Boosting Decision Tree [25] algorithm uses One-hot encoding during the category encoding process. However, when the data dimension is high, the problem of dimension explosion may arise. To address this issue, CatBoost has designed a method called Ordered Target Statistics. This method first randomly arranges all data samples S = {(X_1, Y_1), (X_2, Y_2), (X_3, Y_3), ···, (X_n, Y_n)} to generate multiple sets of random sequences. During the training process, the average label value replaces the category for a particular feature sequence. Assuming that σ = (σ_1, σ_2, σ_3,..., σ_n) is the reordered sequence of the dataset, the *k* feature x_ik_ of the *i* sample in the original dataset can be represented by σ, as shown in Equation (1) [26]. This method can convert categorical features into numerical features, reduce computational complexity, and minimize information loss.
(1)$$x_{\sigma_{p^{k}}}=\frac{\sum_{j=1}^{p-1}\left[x_{\sigma_{j^{k}}}=xM\sigma_{p}k\right]Y_{\sigma j}+aP}{\sum_{j=1}^{p-1}\left[x_{\sigma_{j^{k}}}=\chi_{\sigma_{p}k}\right]+a}$$

After converting categorical features into numerical features using the Ordered Target Statistics method, feature interactions may be affected because numerical features cannot be effectively cross-matched. CatBoost uses a greedy strategy to perform feature interactions. During the first split of the tree generation, CatBoost does not use any cross-features. In subsequent splits, CatBoost uses all of the original features and cross-features that were used to generate the tree, as well as all categorical features in the dataset, to perform feature interactions.

During the model training process using XGBoost and LightGBM algorithms, we found that the model can fit well with F1 wind turbine data, but its fitting effect was poor when testing F2 wind turbine data. Based on this, the CatBoost algorithm proposed the idea of Ordered Boosting, which can effectively reduce the error of gradient estimation and alleviate the problem of prediction shift.

The CatBoost algorithm uses a symmetric binary tree as the base model, and this tree’s structural constraint has a certain regularity effect. For the prediction process of the CatBoost algorithm, the splitting of each feature is independent and not sequential. Multiple samples can be predicted together, improving the prediction speed of the CatBoost algorithm.

### 2.2. Introduction to ASL-CatBoost Algorithm

The icing fault detection of wind turbines is a typical imbalanced data classification problem. During the entire lifecycle of a wind turbine’s operation, fault data only accounts for a very small portion of that operation, which can easily cause the model to be greatly affected by normal data, and it can make it difficult to improve the detection accuracy of fault data. The default cross entropy loss function of the CatBoost algorithm is not good at dealing with the problem of unbalanced positive and negative samples in the dataset. To solve the problem of unbalanced positive and negative samples in the dataset, He Kaiming and others proposed Focal Loss [27], as shown in Equation (2), where *P_t_* is the probability that the prediction sample is a positive sample and *γ* is a weight parameter. However, in the actual scenario application process, the author found that the accuracy of the loss function was not high enough.
(2)$$\mathrm{Focal}\;\mathrm{Loss}=\{\begin{array}{c} -{\left(1-P_{t}\right)}^{\gamma }\log P_{t},y=1\\ -P_{t}{}^{\gamma }\log(1-P_{t}),y=0 \end{array}$$

Therefore, this paper proposes an improved asymmetric loss function based on the focal loss function and considers the application of ASL for the CatBoost algorithm. The main innovations of the asymmetric loss function are as follows:

(1) As shown in Equation (3), the asymmetric loss function focuses on the *γ* Parameter decoupling to *γ*+ and *γ*−. The loss weights of positive samples and negative samples are adjusted using the asymmetric focusing method to reduce the impact of negative samples and simple samples on the loss function and help the model better learn meaningful features in positive samples and difficult to detect samples.
(3)$$ASL*=\{\begin{array}{c} -{\left(1-P_{t}\right)}^{\gamma +}\log P_{t},y=1\\ -P_{t}{}^{\gamma -}\log(1-P_{t}),y=0 \end{array}$$

(2) In order to reduce the contribution of negative samples to the loss function as much as possible, with high confidence, Asymmetric Loss proposed a probability transfer mechanism to process the hard threshold of negative samples with high confidence. As shown in Equation (4), *m* ≥ 0 is an adjustable hyperparameter, which is generally set to 0.2. When the predicted probability *Pt* of positive samples is less than the set hyperparameter m, it indicates that the current sample has a high probability of being a negative sample. Therefore, the probability of predicting the sample as a positive sample can be directly set to 0, and the probability result of predicting the sample as a positive sample can be returned in *P_m_*.
(4)$$P_{m}=MAX\left(P_{t}-m,0\right)$$

After completing the above two improved methods, the Asymmetric Loss expression was finally obtained, as shown in (5).
(5)$$\mathrm{ASL}=\{\begin{array}{l} -{\left(1-P_{t}\right)}^{\gamma +}\log P_{t},\;y=1\\ -{\left(P_{m}\right)}^{\gamma -}\log(1-P_{m}),y=0 \end{array}$$

In conclusion, this paper replaces the default cross entropy loss function of the CatBoost algorithm with Asymmetric Loss, and it proposes the ASL-CatBoost algorithm, which makes the algorithm more sensitive to fault data. The Asymmetric Loss function has three adjustable parameters, namely, $\gamma^{+}$, $\gamma^{-}$, and m. During the process of detecting the icing fault of wind turbines using the ASL CatBoost algorithm, the author found that it is more appropriate to set $\gamma^{+}$ as 2, $\gamma^{-}$ as 3, and *m* as 0.3. The model training was conducted under this super parameter. Since the prediction probability is a value between 0–1, the value after the 3rd power is less than the value after the 2nd power, therefore, the impact of negative samples on the Loss function will be reduced. At the same time, for a negative sample, if the predicted result is 0.1, the confidence level of the negative sample is very high. The Loss function will conclude that it is a negative sample, and the weight influence of the sample on the Loss function is 0. Therefore, the improved ASL CatBoost algorithm will focus on training difficult to detect samples and positive samples to improve the detection accuracy of fault data. The feasibility of this algorithm was verified in Section 4.3 of the article.

### 2.3. Introduction to the Reptile Search Algorithm

Setting the ASL-CatBoost algorithm training hyperparameter, as previously proposed, has a great impact on the accuracy and efficiency of the algorithm’s fault detection abilities. In order to find the optimal parameter combination and reduce the impact of human factors on the accuracy of the algorithm, an improved Reptile Search Algorithm is proposed to optimize the hyperparameter of the ASL-CatBoost algorithm and improve the fault detection speed and detection accuracy of ASL-CatBoost algorithm.

In 2021, Laith Abualigah proposed a meta-heuristic optimizer called the Reptile Search Algorithm (RSA) [28]. The main function of this algorithm is to simulate the hunting behavior of crocodiles. The two main features of crocodile behavior in the algorithm are considered to be ‘rounding up’ and hunting; switching between these two different behaviors is affected by the current number of iterations and the maximum number of iterations. When the current number of iterations is t ≤ T/2, the encirclement strategy is executed; when t > T/2, the hunting phase is performed. The round-up process also includes two steps: high-altitude walking or belly walking. Hunting is achieved through hunting coordination or hunting cooperation. The specific process of the algorithm is as follows:
(1)Initialization phase

In RSA, the optimization process starts with a set of candidate solutions, and in each iteration, the optimal solution obtained is considered to be close to the optimal value. Among them, X is a randomly generated set of candidate solutions, as shown in Equation (6).
(6)$$X_{i,j}=rand\cdot \left(UB-LB\right)+LB,i=1,2,3,\ldots ,N\;j=1,2,3,4,\ldots ,n$$

In the equation, $X_{i,\;j}$ represents the position of the *i*-crocodile individual in the *j* dimension, *N* is the number of candidate solutions, *n* is the dimension of the given problem, rand belongs to the random function in the interval [0, 1], and *LB* and *UB* represent the given lower and upper bounds of the problem.(2)Encirclement stage

When *t* ≤ *T*/2, the algorithm is in the early stage of its iteration, where the crocodile population searches globally and enters the bounding phase. When *t* ≤ *T*/4, the crocodile population adopts a high-altitude walking strategy, and when *T*/4 < *t* ≤ *T*/2, the crocodile population implements an abdominal walking strategy. The position update equation for the crocodile population during the encirclement exploration phase is shown in Equation (7).
(7)$$x_{\left(i,j\right)}\left(t+1\right)=\{\begin{array}{c} Best_{j}\left(t\right)-\eta_{\left(i,j\right)}\left(t\right)\cdot \beta -R_{\left(i,j\right)}\left(t\right)\cdot rand,t\leq \frac{T}{4}\\ Best_{j}\cdot x_{\left(r1,j\right)}\cdot ES\left(t\right)\cdot rand,\;\;\;\;\;\;\;\;\;\;\;\;\;\;\;\frac{T}{4}<t\leq \frac{T}{2} \end{array}$$

In the equation, $Best_{j}\left(t\right)$ represents the position of the optimal solution at the current moment, *t* is the current number of iterations, *T* is the maximum number of iterations, and $\eta_{\left(i,j\right)\;}\left(t\right)$ represents the hunting behavior of the *i* candidate solution in the *j* dimension’s operator; the calculation is shown in Equation (8). *β* is a sensitive parameter which controls the exploration accuracy of the encirclement stage during the iterative process, and it is fixed at 0.1. $R_{\left(i,j\right)}\left(t\right)$ is a reduction function used to reduce the search area value, and it is calculated using Equation (9). r1 is a random integer between (1, N), $x_{\left(r1,j\right)}$, indicating the *j* dimension position of the r1 random candidate solution. *N* is the number of candidate solutions and evolution factor ES(t) is a probability ratio. During the entire iteration process, the value randomly decreased between 2 and −2, and it was calculated using Equation (10).
(8)$$\eta_{\left(i,j\right)}=Best_{j}\left(t\right)\cdot P_{\left(i,j\right)}$$
(9)$$R_{\left(i,j\right)}=\frac{Best_{j}\left(t\right)-x_{\left(r2,j\right)}}{Best_{j}\left(t\right)+\epsilon }$$
(10)$$ES\left(t\right)=2r_{3}\left(1-\frac{t}{T}\right)$$

In the equation, ϵ is a very small positive number, r2 is a random integer of [1, N], $r_{3}$ represents a random integer between [−1, 1], and $P_{\left(i,j\right)}$ represents the percentage difference between the optimal solution and the j dimension position of the current solution, calculated as shown in Equation (11).
(11)$$P_{\left(i,j\right)}=\alpha +\frac{x_{\left(i,j\right)}-M_{\left(x_{i}\right)}}{Best_{j}\left(t\right)\cdot \left(UB_{\left(j\right)}-LB_{\left(j\right)}\right)+\epsilon }$$

$M_{\left(x_{i}\right)}$ represents the average position of the *i* candidate solution, and its calculation is shown in Equation (12). $UB_{\left(j\right)}$ and $LB_{\left(j\right)}$ represent the upper and lower bounds of the j dimensional position, respectively. *α* is a sensitive parameter used to control the search accuracy of hunting cooperation during the iteration process (the difference between candidate solutions), which is fixed to 0.1 in this paper.
(12)$$M_{\left(x_{i}\right)}=\frac{1}{n}\overset{n}{\underset{j=1}{\sum }}x_{\left(i,j\right)}$$

(3)Hunting stage

When *T*/2 < *t*, the population has entered a later stage of iteration, and the crocodile population enters the hunting stage. In this mode, when *T*/2 < *t* ≤ 3*T*/4, crocodiles perform hunting coordination. When 3*T*/4 < *t* ≤ *T*, crocodiles perform hunting cooperation. The relevant equation is shown in Equation (13).
(13)$$x_{\left(i,j\right)}\left(t+1\right)=\{\begin{array}{ll} Best_{j}\left(t\right)\cdot P_{\left(i,j\right)}\left(t\right)\cdot rand, & \frac{T}{2}<\;t\leq \frac{3T}{4}\\ Best_{j}\left(t\right)-\eta_{\left(i,j\right)}\left(t\right)\cdot \epsilon -R_{i,j}\left(t\right)\cdot rand, & \frac{3T}{4}<\;t\leq T \end{array}$$

### 2.4. Improvement Strategy of TtRSA

As mentioned above, the initial positions of RSA crocodile individuals are randomly generated within the search space, and this randomness makes it difficult for the population to obtain a more uniform distribution of initial positions. An uneven distribution of the population may increase the severity of an individual’s blind spot and reduce the population’s diversity. In addition, team cooperation, the search range, and the hunting mechanism of the crocodile population are all updated in terms of the current optimal value, and the individual’s iterative update process lacks mutation mechanisms. If the current optimal individual falls into a local optimum, the population may quickly converge within a short period, resulting in the algorithm being unable to break free from the constraints of the local extreme value. To address the shortcomings of the RSA, this paper considers introducing the Tent chaotic mapping and t-distribution mutation strategy to improve the RSA.

#### 2.4.1. Tent Chaotic Mapping

In response to the problem of uneven population distribution caused by the random initialization of the RSA algorithm, this article introduces Tent chaotic mapping to solve this problem. The Tent chaotic map is a method of implementing chaos control using the tent function as the control function. By introducing the Tent chaotic map to generate pseudo-random numbers to initialize the RSA crocodile population, the traversal of the pseudo-random numbers enables the population to be more evenly distributed throughout the entire search space. This is beneficial for reducing the ‘blind areas’ of crocodile individuals, thus allowing individuals to quickly find better solutions, which improves the convergence speed of the algorithm. Chaotic maps have characteristics such as randomness, traversal, and order, and they can be used to increase the diversity of the population, accelerate the convergence speed of the algorithm in the early stages, and different between chaotic map operators that have different optimization effects. Among them, the Tent chaotic map can produce a uniform chaotic sequence through mapping within the range of (0, 1), and thus, applying the Tent chaotic map to population initialization can increase the diversity of the algorithm population and improve its global optimization ability. The relevant equation is shown in Equation (14).
(14)$$h_{n+1}=f\left(h_{n}\right)=\{\begin{array}{ll} h_{n}/\alpha , & h_{n}\in \left[0,\alpha \right)\\ \left(1+h_{n}\right)/\left(1-\alpha \right), & h_{n}\in \left(\alpha ,1\right] \end{array}$$

In the equation, α ∈ (0,1) is the chaos parameter, $h_{n}$ is a random number within the range of 0 to 1, and n is the chaos variable index. The equation for generating the RSA crocodile population using the Tent chaotic map function is shown in Equation (15).
(15)$$X_{i,j}=h_{n}\cdot \left(UB-LB\right)+LB$$

Figure 1a shows the population distribution based on random initialization, and Figure 1b shows the population distribution based on Tent chaotic mapping. It can be observed that in the two-dimensional space, although the population generated by Tent chaotic mapping does not have the same level of randomness as the population generated by the rand function, the individual position distribution is more uniform and there are no overlapping points or small search blind spots; this can improve population diversity and enable the population to quickly find optimal solutions. The frequency histogram of the population distribution is shown in Figure 2.

#### 2.4.2. t-Distribution Mutation Strategy

The t-distribution is a probability distribution commonly used for parameter estimations and hypothesis testing in situations with small sample sizes. It was proposed by British statistician William Gosset in 1908. The shape of the t-distribution is determined by the degrees of the freedom parameter, where t(n = 1) → N (0,1) and t(n→∞) → C (0,1), where N (0,1) is the normal distribution and C (0,1) is the Cauchy distribution, which are two boundary cases of the t-distribution.

With the development of intelligent optimization algorithms, introducing Gaussian and Cauchy mutations has been proven to effectively improve the algorithm’s ability to search the population and escape local optima. In the early stages of algorithm iteration, the degree of the freedom parameter n is set to a small value, and the t-distribution tends towards the Cauchy distribution, which can effectively increase the diversity of the population and improve the algorithm’s global search ability. As the algorithm iterates during later stages, the degree of the freedom parameter n gradually increases, and the t-distribution tends towards the Gaussian distribution, which narrows the population search range and can effectively improve the algorithm’s ability to explore the local space. In the RSA, the expression for the effect of the t-distribution mutation is shown in Equation (16).
(16)$$X_{news}^{j}=X_{best}^{j}+TD\left(n\right)\cdot X_{best}^{j}$$

The equation can be expressed as follows: $X_{news}^{j}$ is the position of the best solution in the j dimension after adaptive t-distribution mutation perturbation, $X_{best}^{j}$ is the position of the best solution in the j dimension before mutation perturbation, and *TD(n)* represents the t-distribution with a degree of freedom of n.

#### 2.4.3. Summary of TtRSA

In summary of the above, we propose to improve RSA by using the Tent chaos mapping and t-distribution mutation strategy to address the problems of the initialized population of the RSA algorithm; for instance, the population is not uniformly distributed and easily falls into a local optimum during iteration. Based on the above proposed improvements, the TtRSA algorithm was ultimately suggested. Section 3 validated the feasibility of the improvement strategy of the TtRSA algorithm based on 11 benchmark test functions.

## 3. Improved Intelligent Optimization Algorithm Experiments

### 3.1. Experimental Design and Test Functions

Experimental Setup: The experiments were conducted on a computer system with the Windows 11 operating system, AMD R7 5800H 3.2GHz processor, and 16 GB of RAM. MATLAB R2022a was used for conducting the experiments. To evaluate the optimization performance of the improved TtRSA algorithm, 11 benchmark test functions were selected, including both unimodal and multimodal functions, that could evaluate the optimization performance of the algorithm for different types of problems. Among them, functions f1–f5 are continuous unimodal functions that are often used to test the optimization accuracy of search algorithms. Functions f9–f13 are multimodal test functions that can evaluate the convergence speed and accuracy of the algorithm and function f15 is a typical fixed-dimension multimodal function, commonly used to test the algorithm’s ability to escape local optima. The relevant information concerning the benchmark test functions is shown in Table 1.

### 3.2. Improvement Analysis of Optimization Algorithm Performance

In this section, the particle swarm optimization algorithm (PSO) [29], whale optimization algorithm (WOA) [30], chimpanzee optimization algorithm (CHOA) [31], and RSA [32] were used as benchmark algorithms to compare with the optimization performance of the improved TtRSA. The experimental settings included a population size of N = 30, a spatial dimension of D = 30, and a maximum number of iterations of T = 1000. Each algorithm was independently run 30 times with the test functions, and the average result of the 30 runs was taken as the final result.

Based on the comparison results in Table 2, it can be observed that under the same constraints, for the single-peaked test functions f1–f5, the optimization results of the TtRSA were several orders of magnitude (or even several tens of orders of magnitude) higher than those of other improved algorithms. Moreover, f1–f4 were able to converge to the theoretical optimal value of 0. For the complex multi-peaked test functions f9–f13, the optimization results of TtRSA were also better than those of other improved algorithms, and f9 and f11 were able to find the optimal value of 0. For the fixed-dimension multi-peaked test function f15, TtRSA was able to generally converge to the vicinity of the theoretical optimal value. The overall optimization performance of TtRSA was excellent for all 11 benchmark test functions, whether single-peaked test functions or complex multi-peaked test functions. This demonstrates the outstanding stability and robustness of TtRSA, and it proves that the TtRSA algorithm, which integrates multiple strategies, has strong global exploration and local development capabilities.

### 3.3. Convergence Performance Analysis of the Improved Optimization Algorithm

To visually and intuitively compare the convergence of algorithms in the function optimization process, the convergence curves were analyzed, as shown in Figure 3, where the vertical axis represents the fitness value of the corresponding function, and the horizontal axis represents the number of iterations of the optimization algorithm. Figure 3a–e show the running results of five optimization algorithms on a unimodal function. It is evident that the convergence curve of the TtRSA algorithm decreases faster than the other 4 algorithms, whereas the convergence curves of the remaining four algorithms all exhibit varying degrees of stagnation, indicating a lower optimization accuracy. This suggests that applying the improved Tent chaotic mapping strategy to initialize the population increases the diversity of the crocodile population; this makes the initial solution distribution more uniform, and it indicates that the algorithm can find the optimal solution quickly and more easily. Figure 3f–h show the convergence curves of the five optimization algorithms on a multimodal function. It is evident that in the first stage, the TtRSA algorithm’s convergence speed is significantly faster, further demonstrating the effectiveness of the improved Tent chaotic mapping and t-distribution mutation strategy; this changes the crocodile population’s search step and greatly improves the optimization accuracy and speed of the RSA [33].

Regarding the above experiments, it is evident that the proposed method, based on the Tent chaotic mapping and t-distribution mutation strategy, can effectively solve the effects of uneven population distribution and the difficulties in jumping out of local optima during the initialization of the RSA algorithm. The TtRSA algorithm has significant advantages over the RSA and the other four algorithms in terms of optimization precision, convergence speed, and ability to escape from local optima. Section 4.5 verifies the feasibility of optimizing the hyperparameters of the ASL-CatBoost algorithm based on the TtRSA algorithm to achieve ice fault detection in wind turbines.

## 4. Ice Fault Detection Experiment for Wind Turbines

Prognostics and health management are crucial for the lifecycle monitoring of equipment, especially complex equipment such as wind turbines that operate in harsh environments. Improving the speed and accuracy of fan fault detection can reduce maintenance costs and optimize work efficiency. This section aims to verify the effectiveness of the proposed ASL-CatBoost fault detection algorithm and TtRSA with regard to the application of wind turbine fault detection.

### 4.1. Wind Turbine Icing Fault Dataset

The fault detection experiment dataset used in this paper uses the SCADA system data information of two three-bladed wind turbines, F1 and F2, provided by Goldwind, under real operating conditions. There are three state modes in the dataset: icing fault, normal state, and invalid state (wherein it is difficult to determine the type of state). The dataset contains 27 feature dimensions in total, the time span of the F1 wind turbine is two months, and the time span of the F2 wind turbine is one month. The dataset provided the normal operation data and data concerning the specific time period wherein the wind turbine failed due to blade icing. Some dataset examples are shown in Table 3.

The time periods of normal operation and icing faults in the dataset are provided in separate Excel files, as shown in Table 4. Therefore, it is necessary to annotate the dataset based on specific state time periods. The Python ‘append()’ method can be used for annotation, where normal operation data are labeled as 0, and fault operation data are labeled as 1. Data analysis revealed the presence of a few unannotated invalid data points in the dataset, which have unknown operating states. These unannotated data points can negatively impact the accuracy of model training and increase computational overheads. Therefore, they should be removed during the data preprocessing stage.

### 4.2. Evaluating Indicator

Fault detection is a typical binary classification problem. The Confusion matrix is often used to measure the accuracy of the classifier, as shown in Table 5 below. The icing data are 1, and the normal data are 0. TP is the true example, representing both the diagnostic category and the actual category as icing data. FN is a false negative case, representing the diagnostic category, Normal data, and the actual category is icing data. FP is a false positive example, indicating that the diagnostic category is icing data when the actual category is Normal data. TN is a true negative example, indicating that both the diagnostic category and the actual category are normal data.

In accordance with the Confusion matrix, three evaluation indicators are extended: Precision, Recall, and F1 score. As shown in Equation (17), Precision refers to the proportion of the number of correctly predicted fault samples identified by the algorithm to the total number of predicted fault samples. As shown in Equation (18), the recall rate (Recall) refers to the proportion of the number of correctly predicted fault samples identified by the algorithm to the total number of true fault samples. As shown in Equation (19), the F1 score is the harmonic mean of accuracy and recall. The higher the values of the above three indicators, the better the algorithm performance.
(17)Precision=TP/(TP+FP)
(18)Recall=TP/(TP+FN)
(19)$$F1_{score}=\frac{2\times Precision\times Recall}{Precision+Recall}$$

To more intuitively demonstrate the advantages and disadvantages of the algorithm in terms of classification problems, this article also introduces the Receiver Operating Characteristic Curve (ROC) to evaluate the performance of the classification model, as shown in Figure 4. ROC is a curve that visually describes the true positive rate and false positive rate of a classification model based on different thresholds. The horizontal axis of the ROC curve is FPR, which represents the false positive rate, and the vertical axis is TPR, indicating sensitivity. The data points of the ROC curve are calculated using the TPR and FPR values obtained from the classification model at different thresholds. On the ROC curve, it is generally hoped that the curve will be closer to the upper left corner because at this point, the true probability (TPR) is high, whereas the false probability (FPR) is low, indicating that the classification model performs better. Usually, the better the performance of a classifier, the larger the Area Under Curve (AUC) below the ROC. The range of AUC values is 0.5 to 1, where 0.5 represents a completely random classification effect and 1 represents a perfect classifier. Therefore, both ROC and AUC can be used to evaluate the performance of classification models; when ROC is closer to the upper left corner and AUC values are closer to 1, it indicates that the model’s performance is better.

### 4.3. ASL-CatBoost Experiment

This chapter’s experiment aims to demonstrate the effectiveness of the proposed ASL-CatBoost algorithm. During the training process of the wind turbine fault detection algorithm, based on the ASL-CatBoost algorithm, the F1 wind turbine dataset was used as the training set and validation set. In order to prevent the model from overfitting, a 10-fold cross-validation method was used to improve the generalization ability of the model during the training process, and the optimal model was retained after the training was completed. To test the performance of the fault detection algorithm, the F2 wind turbine data was used as the test dataset, which included 10,638 fault data and 168,930 normal data.

This section’s aim is to verify the effectiveness of the improved ASL-CatBoost algorithm. Using the F1 wind turbine icing dataset as the training set, to prevent overfitting of the model, a 10-fold cross validation method was used during the training process to enhance the model’s generalization ability. To test the performance of the fault detection algorithm, the F2 wind turbine data in the second section were used as the test dataset, which included 10,638 fault data and 168,930 normal data. The ASL-CatBoost algorithm model, as well as classic machine learning algorithm models, such as the GBDT and Deep Learning model (LSTMAE), were used for comparative experiments. The training and testing datasets used for each algorithm were the same, and default parameters were used for hyperparameters. The fault detection performance of different algorithms is shown in Table 6. To verify the effectiveness of the two improvement methods of the loss function in this paper, ablation experiments were conducted. The CatBoost^1^ algorithm used the asymmetric focusing strategy of Equation (3) to complete the decoupling of the loss weight γ parameters in the focal loss function, and CatBoost^2^ only refers to Equation (4) for the hard thresholding of negative samples with high confidence. The experimental results show that the two improved strategies exhibit certain improvements, with regard to various evaluation indicators, compared with the initial CatBoost algorithm. Overall, the improved ASL-CatBoost algorithm in this article improved the recall rate by approximately 1% and it improved the accuracy and F1 score by approximately 2%, as compared with the original algorithm. Compared with the LSTMAE model, it improved the accuracy by 9% and the recall rate by 6%. The experimental results validate the feasibility of the improved algorithm.

To more intuitively compare the advantages and disadvantages of each algorithm, Figure 5a–e shows the ROC and AUC values of each algorithm in the wind turbine icing fault detection data validation set. From the figure, it is evident that the improved ASL-CatBoost algorithm exhibits a better fault detection effect than the traditional integrated learning algorithm.

### 4.4. TtRSA Algorithm Optimization ASL-CatBoost Algorithm Introduction

#### 4.4.1. TtRSA Optimized ASL-CatBoost Algorithm Process

The ASL-CatBoost algorithm is greatly affected by hyperparameters, and the artificially set hyperparameter may not achieve the optimal effect during the algorithm training process. Therefore, this section proposes to use the improved TtRSA optimization algorithm to optimize the ASL-CatBoost algorithm with hyperparameters. The specific steps are as follows, and the process is shown in Figure 6.

Step 1: Data preprocessing. There are issues with missing samples and the incomplete labeling of sample labels in the icing fault data of wind turbines. It is necessary to preprocess the dataset to ensure that it meets the training requirements.

Step 2: Dataset partitioning. Divide the preprocessed dataset and determine the training, testing, and validation sets for the ASL-CatBoost algorithm.

Step 3: Set model parameters. Set the crocodile population size N and the maximum number of iterations T. In accordance with Equation (14), the Tent chaotic map is used to randomly initialize the individual positions of the crocodile population, i = 1, 2,..., N. Let the parameter t of the current number of iterations = 1. Set the value range of the maximum number of iterations of the CatBoost algorithm decision tree to (100, 2000). Set the value range of the learning rate to (0, 0.2). Set the value range of l2_leaf_reg to (0, 10) and the depth range of the tree to (0, 16).

Step 4: Calculate the fitness values of all crocodile individuals and save the current optimal crocodile individual position *X_Best_*.

Step 5: Determine whether t ≤ T/2 is true, and if it is true, use Equation (7) to implement the encirclement mechanism. When t ≤ T/4, implement the high-level walking strategy to update the individual crocodile position; when T/4 < t ≤ T/2, implement the abdominal crawling strategy to update the individual crocodile position. If t > T/2, use Equation (13) to implement the hunting mechanism. When T/2 < t ≤ 3T/4, execute the hunting coordination strategy to update the individual position of the crocodile; when 3T/4 < t ≤ T, execute the hunting cooperation mechanism. The policy updates the position of the crocodile individual.

Step 6: Use Equation (16) to perturb the t-distribution mutation strategy on some crocodile individuals and compare the fitness value of the crocodile individual after the updated position with the original individual. Reorder the crocodile individuals in accordance with fitness value and retain the current optimal fitness Degree value *X_Best_*.

Step 7: Set t = t + 1 to judge whether the current termination condition is satisfied, that is, whether the maximum number of iterations of the algorithm Itermax has been reached. If the maximum number of iterations of the algorithm has been reached, the currently saved optimal individual fitness value of the crocodile and the best parameter *X_Best_* are outputted, and the algorithm ends, otherwise, go to step 4.

Step 8: After the cycle ends, the obtained global optimal result, that is, the optimal hyperparameters of the ASL-CatBoost, may be substituted into the algorithm for model training, and the optimization effect may be tested.

#### 4.4.2. Experiment for Optimizing ASL-CatBoost with TtRSA

In this section, the ASL-CatBoost is trained using the hyperparameter that was optimized by TtRSA. The experimental dataset and experimental equipment are the same as those in Section 4.3. Table 7 introduces the ASL-CatBoost algorithm obtained through optimization and the hyperparameter related to the comparison algorithm.

The experimental results of the optimized fault detection algorithm, based on the TtRSA algorithm, are shown in Table 8. From the table, it is evident that the ASL-CatBoost algorithm, optimized using TtRSA, has a better detection accuracy and recall rate throughout the whole dataset than the ASL-CatBoost algorithm with manually set hyperparameters. Regarding Table 6, it is evident that the accuracy and recall of machine learning algorithms (LightGBM, SVM, etc.) and the Deep Learning algorithm (LSTMAE) are improved after TtRSA optimization. The results demonstrate the effectiveness of the hyperparameter search for fault detection algorithms based on the TtRSA algorithm. Moreover, the training time complexity analysis of each algorithm is given in Table 4, which shows that the improved TtRSA-ASL-CatBoost algorithm has significantly less training time than the other algorithms. This is due to the fact that the algorithm obtains a higher accuracy faster at a lower number of iterations after hyperparameter optimization. For fault data detection, each algorithm can detect in time, therefore, the detection speed is negligible.

### 4.5. Enhanced Model Robustness

The robustness of the fault detection algorithm in different scenarios can be improved by adjusting the hyperparameters. To improve the robustness of the model using a different number of features, the model can be trained with 8 features, 16 features, and 22 features extracted from the wind turbine icing fault dataset used in this paper. The method can be used to predict the optimal model parameters in a specific scenario and to improve the robustness and generalization of the model under different scenarios. The optimal hyperparameters and accuracy rates for the three feature count cases are shown in Table 9.

## 5. Conclusions

Icing faults of wind turbines can easily lead to serious economic losses. This paper proposes using the improved ASL-CatBoost algorithm to solve the problem of unbalanced positive and negative samples in the wind turbine fault dataset, and to solve the problem concerning the fault detection algorithm, which is sensitive to the setting of hyperparameters; hence an improved crawler search algorithm is proposed to optimize hyperparameters. The following conclusions can be obtained:(1)Replacing the Cross-entropy Loss function of CatBoost algorithm with the asymmetric Loss function can improve the detection accuracy of the algorithm regarding fault data.(2)The use of the Tent chaotic mapping and t-distribution mutation strategy can improve the problem of imbalanced population distribution during RSA initialization and the tendency to fall into local optima during the iteration process.(3)Optimizing the hyperparameters of the ASL-CatBoost algorithm, based on the TtRSA algorithm, can effectively improve the detection speed and accuracy of the ASL-CatBoost algorithm.

However, this article also has the following limitations; it only details a binary classification problem and it fails to accurately determine which fault is in a multi classification state. In the future, further optimizations should be made to the algorithm to improve the accuracy and efficiency of fault detection. Moreover, future optimizations should enable the algorithm to clearly indicate which category the fault belongs to under multiple fault states. 

## Figures and Tables

**Figure 1 sensors-23-06741-f001:**
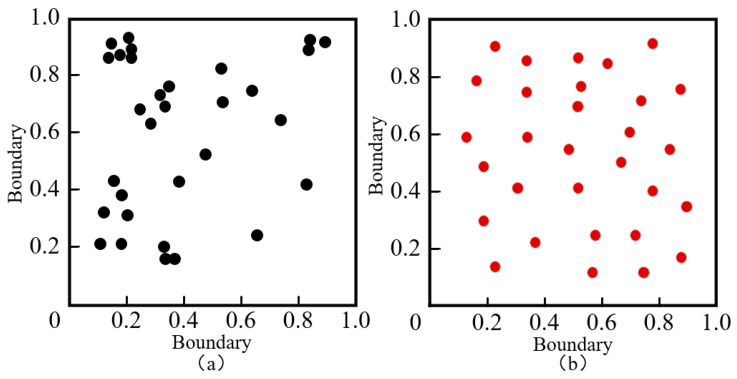
Population initialization distribution diagram. Subgraph (**a**) uses random method for population initialization; subgraph (**b**) uses Tent method for population initialization.

**Figure 2 sensors-23-06741-f002:**
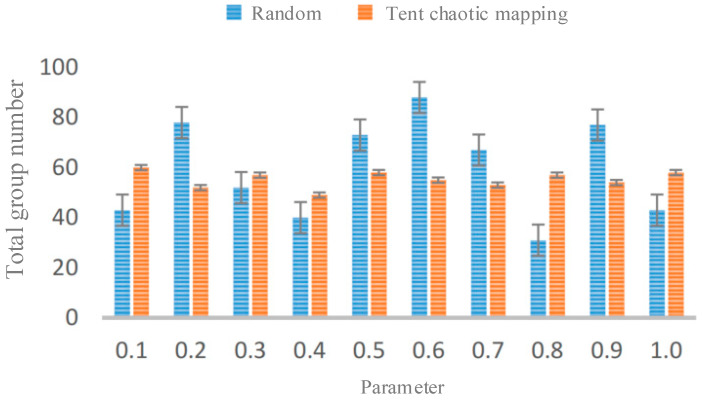
Initialized population frequency histogram.

**Figure 3 sensors-23-06741-f003:**
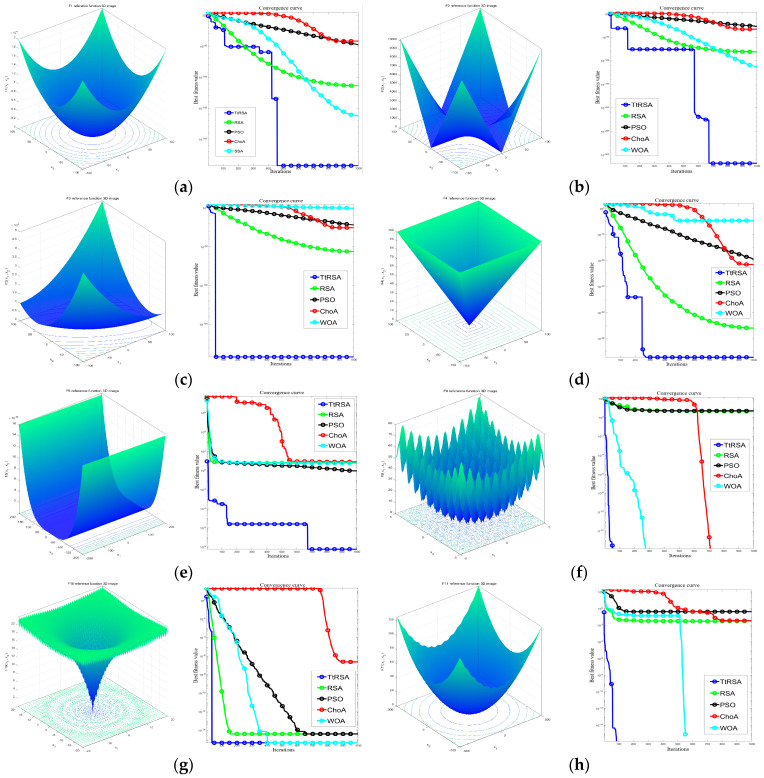
Initial function image and algorithm convergence curve. (**a**) Sphere function convergence curve; (**b**) Schwefel’ 2.21 function convergence curve; (**c**) Schwefel’ 1.2 function convergence curve; (**d**) Schwefel’ 2.21 function convergence curve; (**e**) Rosenbrock function convergence curve; (**f**) Rastrigin function convergence curve; (**g**) Ackley function convergence curve; (**h**) Criewank function convergence curve.

**Figure 4 sensors-23-06741-f004:**
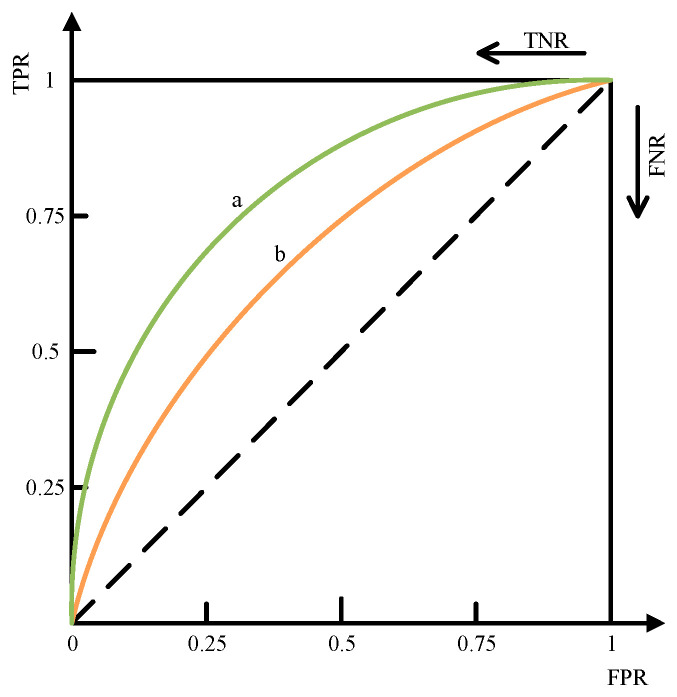
Receiver operating characteristic curve. Curve a and curve b represent the ROC curves under different accuracy rates, and the situation represented by curve a in the figure is better than curve b.

**Figure 5 sensors-23-06741-f005:**
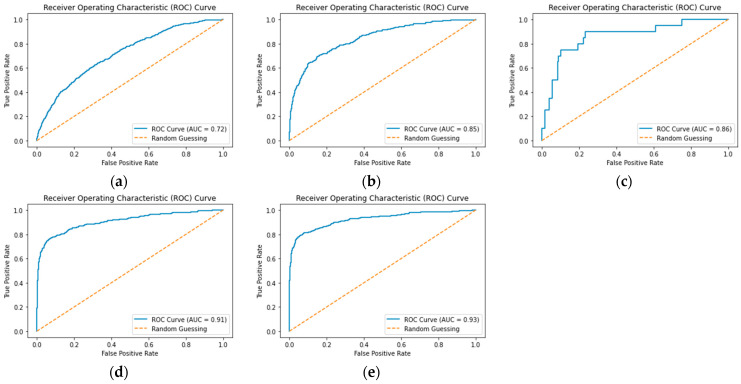
ROC of each algorithm during wind turbine icing fault detection. (**a**) GBDT algorithm Receiver operating characteristics; (**b**) XGBoost algorithm Receiver operating characteristics; (**c**) LightGBM algorithm Receiver operating characteristics; (**d**) CatBoost algorithm Receiver operating characteristics; (**e**) ASL-CatBoost algorithm Receiver operating characteristics.

**Figure 6 sensors-23-06741-f006:**
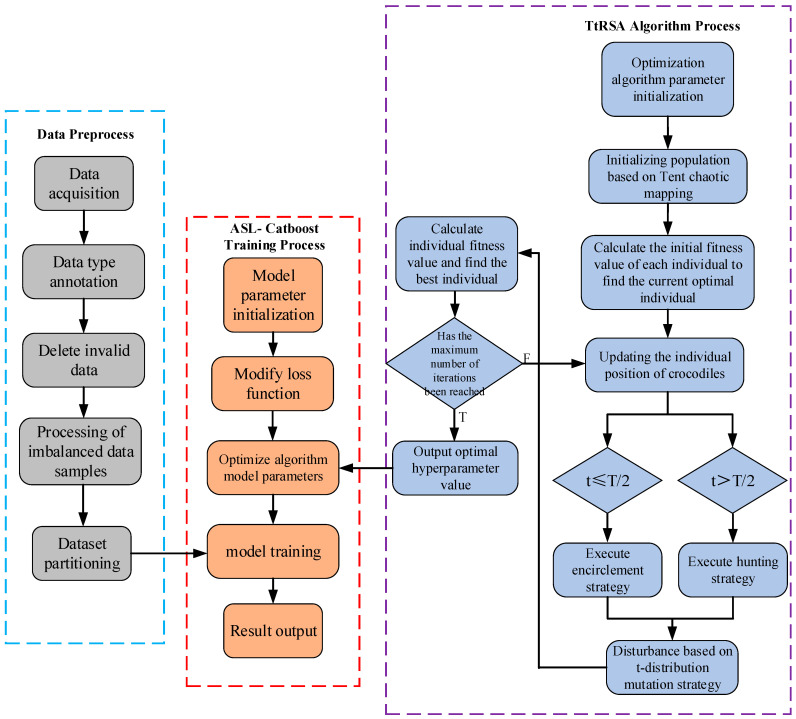
Flowchart of the optimization of the ASL-CatBoost algorithm with TtRSA.

**Table 1 sensors-23-06741-t001:** Benchmark functions.

Number	Test Function	Range	Optima	Type
f1	Sphere	[−100, 100]	0	unimodal
f2	Schwefel’ 2.22	[−10, 10]	0	unimodal
f3	Schwefel’ 1.2	[−100, 100]	0	unimodal
f4	Schwefel’ 2.21	[−100, 100]	0	unimodal
f5	Rosenbrock	[−30, 30]	0	unimodal
f9	Rastrigin	[−5.12, 5.12]	0	multimodal
f10	Ackley	[−32, 32]	0	multimodal
f11	Criewank	[−600, 600]	0	multimodal
f12	Penalized 1	[−50, 50]	0	multimodal
f13	Penalized 2	[−50, 50]	0	multimodal
f15	Kowalik	[−5, 5]	0.0003	multimodal

**Table 2 sensors-23-06741-t002:** Optimization performance analysis of improved optimization algorithms.

Function	PSO		WOA		CHOA		RSA		TtRSA	
	Mean	SD	Mean	SD	Mean	SD	Mean	SD	Mean	SD
f1	1.40 × 10^−30^	2.11 × 10^−32^	2.83 × 10^−164^	1.94 × 10^−163^	1.05 × 10^−30^	1.30 × 10^−20^	0.00	0.00	0.00	0.00
f2	4.21 × 10^−20^	5.54 × 10^−23^	1.52 × 10^−110^	8.28 × 10^−55^	1.24 × 10^−25^	8.55 × 10^−21^	6.88 × 10^−60^	2.58 × 10^−75^	0.00	0.00
f3	7.01 × 10^−18^	2.21 × 10^−21^	6.14 × 10^−2^	1.05 × 10^−5^	6.05 × 10^−19^	1.31 × 10^−17^	5.81 × 10^−42^	4.55 × 10^−73^	0.00	0.00
f4	1.08 × 10^−40^	3.17 × 10^−31^	8.81 × 10^−172^	1.05 × 10^−105^	2.75 × 10^−70^	2.85 × 10^−27^	4.88 × 10^−175^	8.88 × 10^−165^	0.00	0.00
f5	9.67 × 10	6.01 × 10	4.39 × 10^3^	1.05 × 10^5^	3.13 × 10^4^	2.57 × 10^−14^	1.71 × 10	1.37 × 10	2.39 × 10^−20^	1.85 × 10^−21^
f9	4.67 × 10^2^	1.16 × 10	0.00	0.00	1.41 × 10^−01^	1.65 × 10^−26^	6.68 × 10	1.16 × 10	0.00	0.00
f10	2.76 × 10^−16^	5.09 × 10^−21^	9.42 × 10^−37^	1.05 × 10^−5^	1.96 × 10	1.79 × 10^−7^	8.86 × 10^−16^	0.00	7.62 × 10^−78^	5.80 × 10^−56^
f11	1.21 × 10^−1^	7.74 × 10^−3^	1.05 × 10^−18^	7.05 × 10^−25^	4.79 × 10^−02^	5.05 × 10^−18^	9.37 × 10^−2^	7.50 × 10^−1^	0.00	0.00
f12	6.92 × 10^−7^	1.19 × 10^−2^	6.55 × 10^−5^	5.06 × 10^−7^	3.98 × 10^−01^	5.06 × 10^−17^	1.24 × 10^−6^	3.31 × 10^−1^	8.04 × 10^−11^	7.50 × 10^−11^
f13	6.68 × 10^−8^	8.91 × 10^−3^	8.78 × 10^−3^	1.76 × 10^−15^	2.05 × 10^−01^	1.76 × 10^−15^	1.52 × 10^−6^	4.19 × 10^−1^	6.95 × 10^−40^	8.36 × 10^−5^
f15	5.82 × 10^−5^	2.21 × 10^−4^	2.68 × 10^−7^	4.35 × 10^−19^	7.36 × 10^−02^	4.35 × 10^−19^	2.74 × 10^−13^	1.15 × 10^−3^	2.34 × 10^−20^	7.35 × 10^−18^

**Table 3 sensors-23-06741-t003:** Example of a dataset section.

Time	Wind_Speed	Generator_Speed	Power	Wind_Direction	…	Environment_Tmp
2015/11/1 17:33	2.67134589	1.316661063	2.571868051	−0.786603693	…	0.337770344
2015/11/1 17:34	3.058582351	1.293394429	2.537817968	−0.924712235	…	0.337770344
2015/11/1 17:34	3.279860329	1.187032671	2.551855132	−0.962692084	…	0.337770344
2015/11/1 17:34	3.231916767	1.270127794	2.54983978	−0.826309899	…	0.337770344
2015/11/1 17:34	3.364683554	1.329956283	2.557854321	−0.867742461	…	0.337770344
2015/11/1 17:34	3.010638789	1.187032671	2.54983978	−1.157770399	…	0.337770344
2015/11/1 17:34	3.360995587	1.286746819	2.565868862	−1.233730097	…	0.337770344
…	…	…	…	…	…	…
2015/12/1 18:59	1.557580068	1.223594525	1.636697646	1.461112823	…	1.314590648

**Table 4 sensors-23-06741-t004:** Time periods of normal data and fault data.

Fault Operating Time Period		Normal Operating Time Period	
Start Time	End Time	Start Time	End Time
2015/11/4 22:15	2015/11/4 23:33	2015/11/1 17:33	2015/11/4 19:42
2015/11/9 3:21	2015/11/9 5:14	2015/11/5 11:06	2015/11/9 1:23
2015/11/9 21:26	2015/11/9 23:18	2015/11/9 12:20	2015/11/9 19:27
2015/11/13 2:59	2015/11/13 4:51	2015/11/10 12:43	2015/11/13 0:38
2015/11/16 15:31	2015/11/16 15:57	2015/11/13 9:10	2015/11/15 16:35
2015/11/23 20:40	2015/11/23 22:33	2015/11/17 12:14	2015/11/23 18:41
2015/11/24 5:42	2015/11/24 6:31	2015/11/24 1:24	2015/11/24 2:39
2015/11/24 14:58	2015/11/24 16:51	2015/11/24 10:49	2015/11/24 12:12
2015/11/25 20:55	2015/11/25 22:48	2015/11/25 18:00	2015/11/25 18:56
2015/11/26 1:47	2015/11/26 3:40	2015/11/26 10:10	2015/11/28 2:16
2015/11/28 4:15	2015/11/28 6:08	2015/11/28 11:52	2015/11/29 2:30
2015/11/29 4:29	2015/11/29 6:22	2015/11/29 11:48	2015/11/29 14:36
2015/11/29 17:44	2015/11/30 8:52	2015/11/30 10:11	2015/11/30 13:08

**Table 5 sensors-23-06741-t005:** Confusion matrix.

	Icing Diagnosis	Normal Diagnosis
Actual icing	TP	FN
Actual normal	FP	TN

**Table 6 sensors-23-06741-t006:** Performance comparison of each algorithm model.

Models	Precious	Recall	F1-Score	Train Times
GBDT	0.922703	0.936614	0.929606	12 m 16 s
XGBoost	0.907069	0.925287	0.916087	12 m 30 s
LightGBM	0.913742	0.936578	0.925019	13 m 15 s
CatBoost	0.926148	0.930403	0.928271	12 m 11 s
SVM	0.807592	0.735728	0.770886	11 m 20 s
LSTMAE	0.857428	0.886741	0.871838	15 m 30 s
CatBoost^1^	0.934316	0.941628	0.937958	12 m 30 s
CatBoost^2^	0.935743	0.932849	0.934294	12 m 20 s
ASL-CatBoost	0.949427	0.943276	0.946341	12 m 35 s

**Table 7 sensors-23-06741-t007:** Algorithm-related parameters after the optimization of the TtRSA.

Method	Hyper-Parameters
ASL-CatBoost	Iterations = 1000, depth = 6, learning_rate = 0.05, l2_leaf_reg = 0.4
TtRSA-LSTMAE	Hidden_num = 8, windowsize = 100, stride = 1, learning_rate = 0.001, epoch = 16
TtRSA-SVM	c = 40.001, g = 0.008
TtRSA-ASL-CatBoost	Iterations = 300, depth = 8, learning_rate = 0.1, l2_leaf_reg = 0.6
TtRSA-XGBoost	max_depth = 5, min_child_weight = 1, subsample = 0.7, colsample_bytree = 0.8, scale_pos_weight = 1
TtRSA- LightGBM	n_estimators = 144, max_depth = 8, learning_rate = 0.1, random_state = 42, subsample = 0.7, num_leaves = 524

**Table 8 sensors-23-06741-t008:** Comparison of the fault data detection effects of various algorithms.

Models	Precious	Recall	F1-Score	Train Times
ASL-CatBoost	0.949427	0.943276	0.946341	12 m 30 s
TtRSA-SVM	0.837424	0.865792	0.851372	11 m 52 s
TtRSA-LSTMAE	0.882497	0.923769	0.902661	18 m 20 s
TtRSA-LightGBM	0.935598	0.937473	0.936535	15 m 43 s
TtRSA-XGBoost	0.928736	0.918567	0.923624	21 m 17 s
TtRSA-ASL-CatBoost	0.950136	0.949026	0.949581	8 m 52 s

**Table 9 sensors-23-06741-t009:** Comparison of Hyperparameters and Effects of Fault Detection Algorithms under Different Number of Features.

Number of Features	Optimal Case Hyperparameters	Precious	Recall	F1-Score
8	Iterations = 1000, depth = 4, learning_rate = 0.02, l2_leaf_reg = 0.6	0.903581	0.676372	0.758174
16	Iterations = 800, depth = 6, learning_rate = 0.1, l2_leaf_reg = 0.7	0.917795	0.836742	0.975396
22	Iterations = 500, depth = 7, learning_rate = 0.1, l2_leaf_reg = 0.6	0.938331	0.897031	0.917216

## Data Availability

The data are not publicly available as they comprise sensitive information.

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
