# Peer review of "Research on Wind Turbine Fault Detection Based on the Fusion of ASL-CatBoost and TtRSA"

_sensors, 2023, doi:10.3390/s23156741_

Round 1
Reviewer 1 Report
While the paper proposes an algorithm for wind turbine fault detection and highlights the limitations of traditional machine learning algorithms, there are several issues to be solved before the final recommendation:
The introduction does not sufficiently provide the context and motivation for the research. Discussing the significance of wind turbine fault detection, the existing challenges in the field, and the potential impact of improving fault detection accuracy would be beneficial. Refer to some recent papers such as ‘Application of metaheuristic optimization based support vector machine for milling cutter health monitoring’.
Comparison of the proposed model over ML & DL algorithms can be extended by referring to the following articles. Development of Deep Belief Network for Tool Faults Recognition, Application of machine learning for tool condition monitoring in turning
The paper briefly mentions the enhanced ASL-CatBoost and TtRSA algorithms but fails to provide a comprehensive and detailed explanation of these methodologies. Without a thorough understanding of the proposed algorithms, assessing their effectiveness and comparing them to existing techniques is difficult.
The experimental results are mentioned, but there is a lack of detail regarding the datasets used, the evaluation metrics employed, and the methodology for comparing the proposed algorithm with traditional intelligent optimization algorithms and ensemble learning algorithms. Without these details, assessing the validity and generalizability of the results is challenging.
A real picture of the experimental setup and arrangement made for data collection is missing.
The paper does not address the practical implementation challenges of deploying the proposed algorithm in real-world wind turbine systems. Factors such as computational efficiency, scalability, and robustness to varying operational conditions need to be considered and discussed to evaluate the practical viability of the algorithm.
The paper lacks empirical validation on real-world wind turbine systems. While the algorithm is claimed to exhibit notable enhancements in accuracy and recall, there is a need for field testing or at least a more realistic simulation-based validation to support these claims.
How to ensure the robustness of the model in a highly noisy environment? Model augmentation can be an option. You may refer to ‘Augmentation of decision tree model through hyper-parameters tuning for monitoring of cutting tool faults based on vibration signatures.'
The paper does not discuss the limitations of the proposed algorithm or suggest potential future research directions. Identifying limitations would help readers understand the boundaries and applicability of the algorithm, while future directions would highlight opportunities for further improvement and advancement in the field.
Comment on computational time and complexity in the training of the algorithm.
How to deal with the data diversity of the present moment and moment in the future?
How to deal with the possibility of misclassification of a faulty condition as a normal condition (type II error)? If the model is deployed in real-time and such a situation arises, how will you identify that the fault is occurring, however, your system showcased it as normal.
How to deal with the possibility of misclassification of a normal condition as a faulty condition depending on its degree?
I’ve suggested a few of the articles just for your reference and hope that these articles direct and guide you in your future work. If you find them worthy and interesting, you may refer them.
All the best. Looking to receive a revision of your manuscript.
Moderate editing of English language are required
Reviewer 2 Report
1. The introduction section lacks flow and connection in discussion between different paras. Add study background.
2. Few sentences are unnecessarily lengthy and can be divided for better understanding. For instance, in last para of introduction section.
3. One of the indicated existing deficiency in the manuscript is low accuracy and recall rate of the traditional fault detection algorithm. These are not mentioned in related work review. And what is meant by traditional fault detection algorithm? Is it also for recent studies? Add more related recent studies to indicate the research gap and limitations, if exists.
4. The implementation is not well discussed. This needs a careful and major revision to replicate the work.
5. The dataset details need to be discussed. What kind of faults are there? Explain dependent and independent variables clearly.
6. Add key findings in conclusions section.
7. The details of studied models are not given.
8. The developed model is not compared with existing studies. How it is relatively better? Are they trained on same dataset and for similar conditions?
Nil
Reviewer 3 Report
Reviewer’s comments on a manuscript submitted to the MDPI Sensors on “Research on Wind Turbine Fault Detection Based on the Fusion of ASL-CatBoost and TtRSA”
This paper proposed an algorithm based on a hybrid TtRSA-ASL-CatBoost technique to overcome the limitations of the traditional ML algorithms in real-time wind turbine fault diagnosis and classification. The authors also compared the performance of their proposed algorithm with single ASL-CatBoost algorithm and other ensemble learning techniques.
The subject of the paper is very interesting, and it is worthy of consideration. Fault detection is an important topic in the wind energy industry, and given the large volumes of data the offshore wind industry generates on a daily basis, the use of ML techniques has become increasingly popular. Nevertheless, there are some major comments which the reviewer would like to encourage authors to consider before the paper can proceed further:
- The literature review is not comprehensive. Having a dedicated section and reviewing some of the existing ML techniques in fault diagnosis of wind turbines will be very useful.
- Section 2 provides pretty well-known materials and hence it can be shortened. Some content is taken from the literature and therefore they will require references (such as the equations).
- Section 3 and 4 are the most interesting parts of the paper, and they must be expanded. Making comparisons with some of the ML techniques reported in the literature will be very helpful.
- The authors have tested their algorithm on blade icing phenomenon. Given this fault can be very accurately detected by existing icing sensors in industry, it will be useful to support the research by another case study from mechanical/structural faults.
- The results are well presented; however the discussion part is not so strong. The interpretation of the results is very key. Some suggestions on how the performance of the proposed algorithm can be improved will be useful.
There are some typos and grammatical errors which should be looked after.
Round 2
Reviewer 1 Report
Accept
Reviewer 2 Report
The paper can be accepted in present form
Nil